# Association of Funisitis with Short-Term Outcomes of Prematurity: A Frequentist and Bayesian Meta-Analysis

**DOI:** 10.3390/antiox12020534

**Published:** 2023-02-20

**Authors:** Tamara Maria Hundscheid, Maurice Jacob Huizing, Eduardo Villamor-Martinez, František Bartoš, Eduardo Villamor

**Affiliations:** 1Department of Pediatrics, University Medical Center (MUMC+), School for Oncology and Reproduction (GROW), Maastricht University, 6202 AZ Maastricht, The Netherlands; 2Statistics Netherlands, 6412 HX Heerlen, The Netherlands; 3Department of Psychology, University of Amsterdam, 1001 NK Amsterdam, The Netherlands

**Keywords:** funisitis, chorioamnionitis, intrauterine infection, complications of preterm birth, bronchopulmonary dysplasia, retinopathy of prematurity, intraventricular hemorrhage, periventricular leukomalacia, sepsis, mortality

## Abstract

The fetal systemic inflammatory response associated with intra-amniotic inflammation may play a key role in the pathogenesis of complications of preterm birth. Funisitis is the histologic equivalent of the fetal inflammatory response, whereas chorioamnionitis represents a maternal inflammatory response. We conducted a frequentist and Bayesian model average (BMA) meta-analysis of studies investigating the effects of funisitis on short-term outcomes of prematurity. Thirty-three studies (12,237 infants with gestational age ≤ 34 weeks) were included. Frequentist meta-analysis showed that funisitis was associated with an increased risk of any bronchopulmonary dysplasia (BPD), moderate/severe BPD, retinopathy of prematurity (ROP), intraventricular hemorrhage (IVH), periventricular leukomalacia (PVL), any sepsis, early-onset sepsis (EOS), and mortality. However, Bayesian meta-analysis showed that the evidence in favor of the alternative hypothesis (i.e., funisitis is associated with an increased risk of developing the outcome) was strong for any IVH, moderate for severe IVH and EOS, and weak for the other outcomes. When the control group was restricted to infants having chorioamnionitis without funisitis, the only outcome associated with funisitis was any IVH. In conclusion, our data suggest that the presence of funisitis does not add an additional risk to preterm birth when compared to chorioamnionitis in the absence of fetal inflammatory response.

## 1. Introduction

Preterm birth is a public health problem worldwide, with high rates of infant morbidity and mortality [1,2,3]. Intrauterine infection and/or inflammation are major contributors to the complications of very and extremely preterm birth (i.e., below 32 weeks of gestation) [4,5]. However, discerning between the part of the damage that is due to the infectious insult and the part related to the prenatal infection as a primary inducer of prematurity is a conundrum that continues to challenge perinatologists [4].

Invasion by microorganisms and subsequent intra-amniotic inflammation is generally ascending and progressive [4,6,7]. Histopathological chorioamnionitis is defined as diffuse infiltration of neutrophils into the chorioamniotic membranes and it is referred to as acute villitis if the villous tree is affected [4,6,7]. With progression of inflammation, immune cells infiltrate the umbilical cord (umbilical vessels and Wharton’s jelly), resulting in funisitis [4,6,7,8,9,10]. Therefore, whereas chorioamnionitis represents a maternal inflammatory response, funisitis is widely regarded as a fetal response and is considered the histological equivalent of the fetal inflammatory response syndrome [4,6,7,8,9,10].

The systemic inflammatory response is an initially adaptive mechanism of the fetus to infection, but, when unregulated, it can lead to increased injury in the preterm infant [11,12]. Oxidative stress plays a central role in this damage as the inflammatory process can cause a redox imbalance, increasing the release of free radicals and consuming antioxidant defenses [11,12]. Furthermore, very preterm newborns are particularly susceptible to oxidative stress because their antioxidant defenses are significantly lower compared to term infants [13,14]. In fact, oxidative stress is a major contributor to the pathogenesis of adverse outcomes of prematurity, including retinopathy of prematurity (ROP), necrotizing enterocolitis (NEC), bronchopulmonary dysplasia (BPD), intraventricular hemorrhage (IVH), periventricular leukomalacia (PVL), chronic neurodevelopmental and cognitive impairment, and death [13,14]. This has led to the grouping of most of the complications of very preterm birth under the umbrella term oxidative stress disease of prematurity [13,14].

In recent years, numerous observational studies and meta-analyses have investigated the association between chorioamnionitis and complications of prematurity. Some of these studies report data on funisitis and isolated chorioamnionitis, i.e., without funisitis. We hypothesized that the neonatal damage induced by the fetal inflammatory response (funisitis) is more severe than that produced when the inflammatory process is limited to the mother (chorioamnionitis without funisitis). We conducted a systematic review and meta-analysis to test this hypothesis. In addition, we analyzed possible differences in baseline characteristics, such as gestational age (GA), birth weight (BW), sex, or exposure to antenatal corticosteroids, between infants with and without funisitis.

## 2. Patients and Methods

The methodology of this study is based on that of earlier studies of our group on chorioamnionitis and complications of prematurity [15,16,17,18,19]. The study was performed and reported according to the Guidelines for Meta-Analyses and Systematic Reviews of Observational Studies (MOOSE). Review protocol was registered in the PROSPERO international register of systematic reviews (ID = CRD42020203205). The research question was “Is funisitis a risk factor for developing short-term complications (mortality, BPD, IVH, PVL, PDA, NEC, ROP, sepsis) in preterm infants compared to preterm infants without exposure to funisitis?”.

### 2.1. Sources and Search Strategy

A comprehensive literature search was undertaken using the PubMed and EMBASE databases. The search terms involved various combinations of the following key words: funisitis, fetal inflammation, fetal inflammatory response, chorioamnionitis, intrauterine infection, intrauterine inflammation, antenatal infection, antenatal inflammation, bronchopulmonary dysplasia, chronic lung disease, retinopathy of prematurity, necrotizing enterocolitis, intraventricular hemorrhage, periventricular leukomalacia, patent ductus arteriosus, sepsis, mortality, risk factors, outcome, cohort, and case-control. No language limit was applied. The literature search was updated up to March 2022. Narrative reviews, systematic reviews, case reports, letters, editorials, and commentaries were excluded, but read to identify potential additional studies. Additional strategies to identify studies included manual review of reference lists from key articles that fulfilled our eligibility criteria, use of “related articles” feature in PubMed, and use of the “cited by” tool in Web of Science and Google scholar.

### 2.2. Study Selection

The current study included observational studies reporting on the effects of funisitis in very and/or extreme preterm infants. Although the GA cutoff for this category of preterm infants is 32 weeks, studies that had an upper inclusion cutoff of 34 weeks were also included. Subsequently, a sensitivity analysis was performed to evaluate the effect of extending the inclusion limit to 34 weeks GA. Studies that included late preterm infants (GA ≥ 34 weeks) or that combined preterm and term infants were excluded. The primary outcome was short-term complications of prematurity, including mortality during first hospital admission, BPD, IVH, PVL, PDA, NEC, ROP, and sepsis.

### 2.3. Data Extraction, Definitions, and Risk of Bias Assessment

Data extracted included citation information, location of the research group, time period of study, study objectives, study design, inclusion and exclusion criteria, neonatal and maternal characteristics, data on funisitis, data on chorioamnionitis, and data on the different complications.

The main meta-analysis compared infants with funisitis (Fun+) vs. infants without funisitis (Fun−). In subsequent meta-analyses, the Fun− group was divided into two subgroups: (1) infants without funisitis but with chorioamnionitis (Fun−/CA+); and (2) infants who had neither funisitis nor chorioamnionitis (Fun−/CA−). Outcomes were categorized as follows: any BPD (defined as requiring >21% oxygen at the postnatal age of 28 days), moderate/severe BPD (defined as requiring >21% oxygen at the postmenstrual age of 36 weeks), severe BPD (defined as requiring >30% oxygen or mechanical ventilation at the postmenstrual age of 36 weeks), any ROP, severe ROP (type 1 pre-threshold ROP or ROP requiring treatment), any PDA, PDA requiring medical or surgical treatment, PDA requiring surgery, any PVL, cystic PVL, any NEC, NEC ≥ stage 2, any IVH (grade 1–4), severe IVH (grade ≥ 3), any sepsis, early-onset sepsis (EOS, defined as sepsis within 72 h of life), and late-onset sepsis (LOS, defined as sepsis after 72 h of life).

Risk of bias was assessed using the Newcastle–Ottawa Scale for cohort or case-control studies [20]. This scale assigns a maximum of 9 points (4 for selection, 2 for comparability, and 3 for exposure/outcome). Newcastle–Ottawa Scale scores ≥ 7 were considered as indicative of low risk of bias and scores of 5 to 6 as indicative of moderate risk of bias.

### 2.4. Statistical Analysis

#### 2.4.1. Frequentist Meta-Analysis

Studies were combined and analyzed using COMPREHENSIVE META-ANALYSIS V3.0 software (Biostat Inc., Englewood, NJ, USA) [20]. Due to anticipated heterogeneity, summary statistics were calculated with a random-effects model. This model accounts for variability between studies as well as within studies [20,21]. Subgroup analyses were conducted according to the mixed-effects model. In this model, a random-effects model is used to combine studies within each subgroup, and a fixed-effect model is used to combine subgroups and yield the overall effect. The study-to-study variance (tau-squared) is not assumed to be the same for all subgroups. This value is computed within subgroups and not pooled across subgroups [20,21].

For dichotomous outcomes, the odds ratio (OR) with 95% confidence interval (CI) was calculated. For continuous outcomes, the mean difference (MD) or Hedges’ g with 95% CI were calculated. When studies reported continuous variables as median and range or interquartile range, we estimated the mean and standard deviation using the method of Wan et al. and the calculator they provided [22]. Statistical heterogeneity was assessed by Cochran’s Q statistic and by the I^2^ statistic. I^2^ was interpreted on the basis of Higgins and Thompson criteria, where 25%, 50%, and 75% correspond to low, moderate, and high heterogeneity, respectively [23]. Potential sources of heterogeneity were assessed through subgroup analysis and/or random-effects (method of moments) univariate meta-regression analysis, as previously described [24]. For continuous covariates (examples: difference in mean gestational age between infants exposed and unexposed to funisitis), we used meta-regression analyses to test whether there was a significant relationship between the covariate and effect size, as indicated by a Z-value and an associated *p*-value. Meta-regression coefficient indicates the change in the log of the OR of the association between mortality and the corresponding exposure for a unit change in the predictor covariate. Subgroups were compared using meta-regression for categorical covariates. For both categorical and continuous covariates, the R^2^ analog, defined as the total between-study variance explained by the moderator, was calculated based on the meta-regression matrix. We used the Egger’s regression test and funnel plots to assess publication bias. Subgroup analyses, meta-regression, and publication bias assessment were performed only when there were at least ten studies in the meta-analysis. A probability value of less than 0.05 (0.10 for heterogeneity) was considered statistically significant.

#### 2.4.2. Bayesian Model Average Meta-Analysis

The results were further supplemented by a Bayesian model average (BMA) meta-analysis [25,26]. BMA employs Bayes factors (BFs) and Bayesian model averaging to evaluate the likelihood of the data under the combination of models assuming the presence vs. the absence of the meta-analytic effect and heterogeneity [25,26]. The BF_10_ is the ratio of the probability of the data under the alternative hypothesis (H_1_) over the probability of the data under the null hypothesis (H_0_). The BF_10_ was interpreted using the evidence categories suggested by Lee and Wagenmakers [27]: <1/100 = extreme evidence for H_0_; from 1/100 to <1/30 = very strong evidence for H_0_; from 1/30 to <1/10 = strong evidence for H_0_; from 1/10 to <1/3 = moderate evidence for H_0_; from 1/3 to <1 weak/inconclusive evidence for H_0_; from 1 to 3 = weak/inconclusive evidence for H_1_; from >3 to 10 = moderate evidence for H_1_; from >10 to 30 = strong evidence for H_1_; from >30 to 100 = very strong evidence for H_1_; and >100 = extreme evidence for H_1_. Consequently, BMA allows us to distinguish the absence of evidence from the evidence of absence [28]. The BFrf is the ratio of the probability of the data under the random-effects model over the probability of the data under the fixed-effects model. The BFrf was interpreted in the following way: <1/100 = extreme evidence for fixed effects; from 1/100 to <1/30 = very strong evidence for fixed effects; from 1/30 to <1/10 = strong evidence for fixed effects; from 1/10 to <1/3 = moderate evidence for fixed effects; from 1/3 to <1 weak/inconclusive evidence for fixed effects; from 1 to 3 = weak/inconclusive evidence for random effects; from >3 to 10 = moderate evidence for random effects; from >10 to 30 = strong evidence for random effects; from >30 to 100 = very strong evidence for random effects; and >100 = extreme evidence for random effects. We used the empirical prior distributions based on the Cochrane Database of Systematic Reviews transformed to logOR; logOR ~ Student-t (µ = 0, σ = 0.78, ν = 5), tau = Inverse-Gamma (k = 1.71, θ = 0.73) [25,26]. We performed the BMA in JASP [29], which utilizes the metaBMA R package [30].

## 3. Results

### 3.1. Description of Studies and Quality Assessment

The PRISMA flow diagram of the search process is shown in Appendix A. Of 325 potentially relevant studies, 33 were included [31,32,33,34,35,36,37,38,39,40,41,42,43,44,45,46,47,48,49,50,51,52,53,54,55,56,57,58,59,60,61,62,63]. These studies included 12,237 infants. Characteristics of the studies are summarized in Appendix A. Risk of bias assessment according to the Newcastle–Ottawa Scale is depicted in Appendix A. All studies received a score of at least six points, indicating a low to moderate risk of bias.

### 3.2. Main Meta-Analyses

Frequentist meta-analyses on the association between funisitis and short-term complications of prematurity (Fun+ vs. Fun−) are summarized in Figure 1. These meta-analyses showed an association between funisitis and odds of developing any BPD (OR 1.45, CI 1.11–1.90), moderate/severe BPD (OR 1.51, CI 1.02–2.23), BPD or death (OR 1.35, CI 1.04–1.76), any ROP (OR 1.59, CI 1.05–2.42), any PVL (OR 1.99, CI 1.07–3.72), any IVH (OR 2.06, CI 1.47–2.89), severe IVH (OR 1.79, CI 1.30–2.46), any sepsis (OR 1.32, CI 1.00–1.73), EOS (OR 2.33, CI 1.15–4.72), and death before discharge (OR 1.58, CI 1.03–2.41). In contrast, funisitis was not significantly associated with severe BPD, severe ROP, PDA, cystic PVL, NEC, or LOS. As shown in Appendix A, exclusion of the seven studies that included infants with GA above 32 weeks but below 34 weeks [54,55,57,58,59,60,62] did not significantly affect the results of the meta-analysis.

As shown in Table 1, BMA analysis demonstrated that in the Fun+ vs. Fun− comparison, the evidence in favor of the alternative hypothesis was strong (BF_10_ from >10 to 30) for any IVH; moderate (BF_10_ from >3 to 10) for severe IVH and EOS; and weak (BF10 from 1 to 3) for mortality, any BPD, moderate/severe BPD, any ROP, severe ROP, PDA requiring treatment, PDA requiring surgery, any PVL, and any sepsis. Conversely, the evidence in favor of the null hypothesis was moderate (BF_10_ from 1/10 to <1/3) for cystic PVL and weak (BF_10_ from 1/3 to <1) for severe BPD, BPD or death, any PDA, any NEC, NEC ≥ stage 2, NEC or death, and LOS. Data on heterogeneity of BMA analysis are depicted in Appendix A.

When the control group included infants who had neither funisitis nor chorioamnionitis (Fun−/CA− group), frequentist meta-analysis showed an association between funisitis and odds of developing any BPD, moderate/severe BPD, any ROP, any IVH, severe IVH, any sepsis, EOS, and death before discharge. In contrast, frequentist meta-analysis could not demonstrate a significant association for severe BPD, BPD or death, severe ROP, PDA, NEC, PVL, or LOS (Figure 2 and Table 2). BMA analysis of this comparison (Fun+ vs. Fun−/CA−) showed that the evidence in favor of the alternative hypothesis was strong (BF_10_ from >10 to 30) for any PBD, any IVH, and severe IVH; moderate (BF_10_ from >3 to 10) for mortality, any ROP, PDA requiring surgery, and EOS; and weak (BF_10_ from 1 to 3) for moderate/severe BPD, BPD or death, severe ROP, any PDA, and PDA requiring medical treatment. Conversely, the evidence in favor of the null hypothesis was moderate (BF_10_ from 1/10 to <1/3) for cystic PVL and weak (BF_10_ from 1/3 to <1) for severe BPD, any PVL, any NEC, NEC ≥ stage 2, NEC or death, and LOS (Table 1).

When the control group included infants without funisitis but with chorioamnionitis (Fun−/CA+ group), the only outcome for which the positive association was maintained in the frequentist meta-analysis was any IVH (OR 1.58, CI 1.06–2.36) (Figure 2 and Table 2). BMA analysis showed that the evidence in favor of the alternative hypothesis was weak (BF_10_ from 1 to 3) for any IVH and NEC or death. Conversely, the evidence in favor of the null hypothesis was moderate (BF_10_ from 1/10 to <1/3) for mortality, BPD or death, PDA requiring medical treatment, PDA requiring surgery, severe IVH, and LOS; and weak for any BPD, moderate/severe BPD, severe BPD, any ROP, severe ROP, any PDA, any PVL, cystic PVL, any NEC, NEC ≥ stage 2, any sepsis, and EOS (Table 1).

Finally, we also conducted a meta-analysis including the three studies that differentiated grades or severities of funisitis [47,56,61]. As shown in Appendix A, this meta-analysis did not demonstrate an effect of funisitis grade on any of the outcomes.

Neither visual inspection of funnel plots (Appendix A) nor Egger’s test suggested publication or selection bias for any of the meta-analyses that include ten or more studies. Publication bias was not analyzed for meta-analyses including less than ten studies.

**Figure 2 antioxidants-12-00534-f002:**
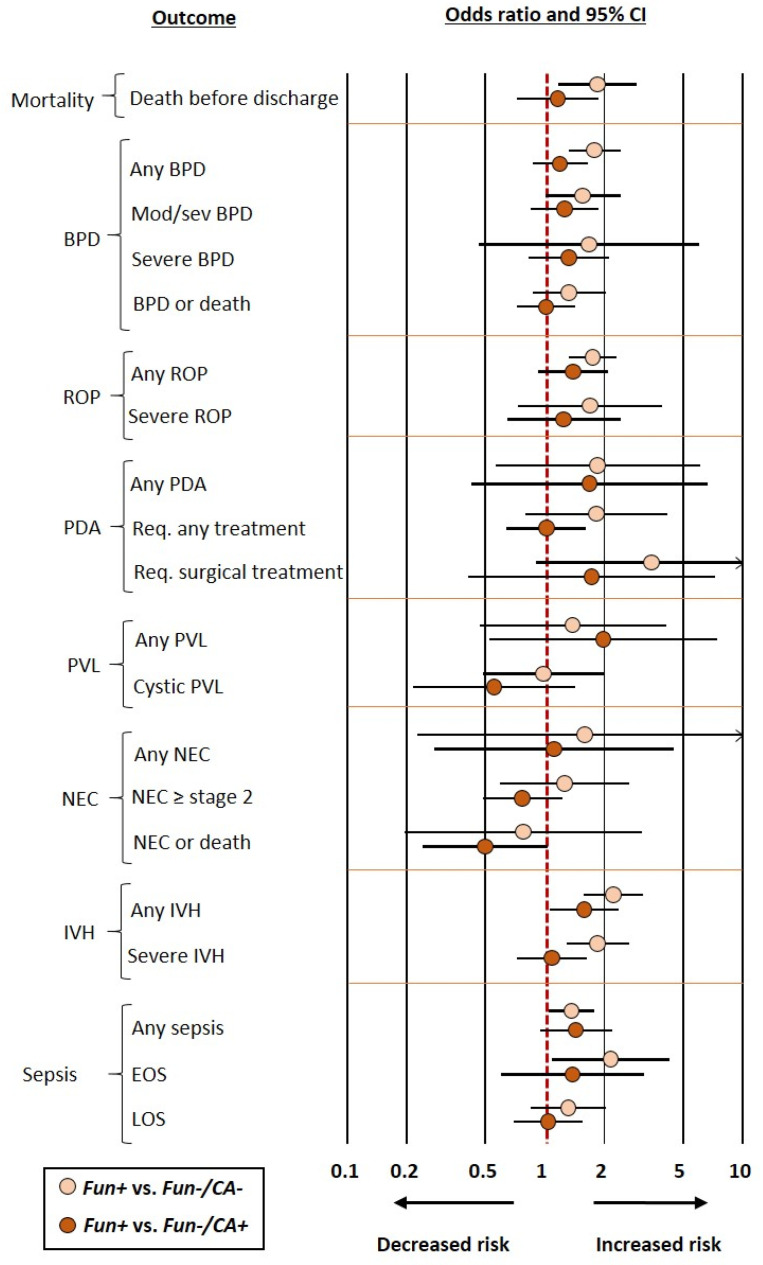
Summary of frequentist random-effects meta-analyses comparing infants with funisitis (Fun+) vs. infants who had neither funisitis nor chorioamnionitis (Fun−/CA−) or infants without funisitis but with chorioamnionitis (Fun−/CA+). BPD: bronchopulmonary dysplasia; EOS: early-onset sepsis; IVH: intraventricular hemorrhage; K: number of studies included in the meta-analysis; LOS: late-onset sepsis; NEC: necrotizing enterocolitis; PDA: patent ductus arteriosus; PVL: periventricular leukomalacia; Req.: requiring, ROP: retinopathy of prematurity.

### 3.3. Additional Meta-Analyses and Meta-Regression

We conducted additional meta-analyses to investigate possible differences in baseline characteristics between infants with and without funisitis. As shown in Table 3, infants in the Fun+ group had significantly lower GA and BW than infants in the Fun− group. In addition, infants in the Fun+ group were more frequently female, were less frequently small for GA or had fetal growth restriction, were less frequently exposed to hypertensive disorders of pregnancy, and were less frequently born by cesarean section than infants in the Fun− group. All these significant differences were maintained when the Fun+ group was compared with the Fun−/CA− group (Table 3). In addition, infants in the Fun−/CA− group showed a lower rate of exposure to antenatal corticosteroids than infants in the Fun+ group (Table 3). In contrast, when the Fun+ group was compared with the Fun−/CA+ group, only the association with sex remained significant. That is, the Fun+ group included significantly more females than the Fun−/CA+ group (Table 3). In addition, three studies reported interleukin (IL)-6 levels in cord blood of preterm infants with funisitis. These levels were markedly elevated in the Fun+ group when compared to the Fun− group, the Fun−/CA+ group, or the Fun−/CA− group (Appendix A).

As described in our previous meta-analyses on chorioamnionitis [15,16,17,18,19], we conducted a meta-regression analysis to investigate the possible correlation between the difference in GA between the Fun+ group and the Fun− group and the effect size of the association between funisitis and outcome. This meta-regression analysis was only performed for the three outcomes (mortality, moderate/severe BPD, and any sepsis) that were reported in ten or more studies. As shown in Appendix A, none of these meta-regressions showed statistically significant results. In addition, the meta-regression also showed no significant correlation between the effect size of the association between funisitis–outcome and sex differences or differences in rate of exposure to antenatal corticosteroids (Appendix A).

**Table 3 antioxidants-12-00534-t003:** Meta-analyses for other covariates.

Control Group	Meta-Analysis	K	Effect Size	95% CI		*p*	Heterogeneity	
				Lower Limit	Upper Limit		I^2^ (%)	*p*
Absence of funisitis (Fun−)	GA (MD in weeks)	20	−1.608	−2.631	−0.586	0.002	99.0	<0.001
	Birth weight (MD in grams)	20	−101.6	−164.6	−38.6	0.002	87.8	<0.001
	Male sex (OR)	19	0.822	0.727	0.928	0.002	4.1	0.407
	Antenatal corticosteroids (OR)	17	1.190	0.905	1.563	0.213	57.7	0.002
	Caesarean delivery (OR)	12	0.506	0.346	0.739	<0.001	81.0	<0.001
	HDP (OR)	11	0.077	0.035	0.168	<0.001	76.2	<0.001
	SGA/IUGR (OR)	8	0.195	0.067	0.564	0.003	86.3	<0.001
Absence of funisitis and chorioamnionitis (Fun−/CA−)	GA (MD in weeks)	17	−1.374	−1.931	−0.817	<0.001	95.5	<0.001
	Birth weight (MD in grams)	15	−76.2	−163.6	11.2	0.087	91.1	<0.001
	Male sex (OR)	17	0.825	0.723	0.942	0.005	4.1	0.406
	Antenatal corticosteroids (OR)	14	1.420	1.098	1.837	0.008	44.2	0.038
	Caesarean delivery (OR)	12	0.378	0.232	0.616	<0.001	87.1	<0.001
	HDP (OR)	10	0.030	0.008	0.112	<0.001	89.7	<0.001
	SGA/IUGR (OR)	8	0.111	0.017	0.712	0.020	95.3	<0.001
Absence of funisitis but presence of chorioamnionitis (Fun−/CA+)	GA (MD in weeks)	15	−0.251	−0.525	0.023	0.072	50.0	0.014
	Birth weight (MD in grams)	15	−25.4	−71.4	20.7	0.281	43.1	0.039
	Male sex (OR)	17	0.800	0.677	0.945	0.009	0.0	0.931
	Antenatal corticosteroids (OR)	15	0.955	0.703	1.298	0.768	30.2	0.129
	Caesarean delivery (OR)	12	0.944	0.754	1.181	0.612	8.4	0.363
	HDP (OR)	10	0.406	0.185	0.894	0.025	50.8	0.032
	SGA/IUGR (OR)	8	0.769	0.445	1.329	0.347	4.0	0.399

Fun−: absence of funisitis regardless of the state of chorioamnionitis; Fun−/CA−: absence of funisitis and chorioamnionitis; Fun−/CA+: absence of funisitis but presence of chorioamnionitis; GA: gestational age; HDP: hypertensive disorders of pregnancy; K: number of studies; MD: mean difference (funisitis minus control group); OR: odds ratio; SGA/IUGR: small-for-gestational age/intrauterine growth restriction.

## 4. Discussion

To the best of our knowledge, this the most extensive and comprehensive systematic review and meta-analysis that investigated the impact of funisitis, as a proxy of the fetal inflammatory response, on the risk of developing short-term complications of very preterm birth. Frequentist meta-analysis showed that funisitis was associated with an increased risk of any BPD, moderate/severe BPD, ROP, IVH, any PVL, any sepsis, EOS, and death before discharge. However, Bayesian meta-analysis showed that the evidence in favor of the alternative hypothesis (i.e., funisitis is associated with increased risk of developing the outcome) was strong for any IVH, moderate for severe IVH and EOS, and weak for the other outcomes that were significant (*p* < 0.05) in the frequentist meta-analysis. Interestingly, when the control group consisted of infants exposed to maternal inflammation but without fetal inflammation (i.e., chorioamnionitis without funisitis), the only outcome that remained significantly associated with funisitis in the frequentist meta-analysis was any IVH. Moreover, the Bayesian meta-analysis showed that the evidence in favor of this association was weak (BF_10_ = 1.9). In addition, the Bayesian evidence in favor of an association between funisitis and complications, such as any BPD, severe IVH, any ROP, PDA requiring surgery, and EOS, becomes moderate-to-strong only when the control group consists of infants with neither fetal (i.e., funisitis) nor maternal (i.e., chorioamnionitis) inflammation. Therefore, our data suggest that the fetal inflammatory response does not add an additional risk to chorioamnionitis for most of the complications of prematurity. We speculate that that a significant part of the pathogenic effects of intrauterine infection/inflammation would be mediated by its role as a trigger of prematurity more than by the alterations induced in perinatal homeostasis.

The main limitation of our meta-analysis is the low number of studies reporting useful data to answer the research question. This is a very common problem in meta-analysis. In fact, a study published in 2011 showed that the median number of trials included in the meta-analyses from the Cochrane Database of Systematic Reviews was three, with an interquartile range from two to six [64]. Bayesian meta-analysis is increasingly being used to address the small-sample challenge [25,26]. In contrast to meta-analysis using frequentist statistics, Bayesian meta-analysis has the ability to quantify evidence in favor or against any hypothesis (including the null hypothesis), and to discriminate, therefore, between absence of evidence and evidence of absence [25,26,65,66]. This is a relevant advantage over the dichotomous interpretation of frequentist inference (significant vs. non-significant), particularly when *p*-values are close to the conventionally accepted limits (i.e., *p* < 0.05) [26,66,67]. This is clearly illustrated in Table 1 with the example of “any IVH”. When the control group consisted of infants without funisitis or chorioamnionitis (Fun−/CA−), the association between funisitis and any IVH in the frequentist analysis was “highly significant” (*p* < 0.001). When the control group consisted of infants without funisitis but with chorioamnionitis (Fun−/CA+), the *p*-value (=0.026) of the frequentist meta-analysis was higher but still significant. By evaluation of BF_10_-values, Bayesian meta-analysis allows us to interpret that when the Fun−/CA− group was the control, the data were 65.6 times more likely under the presence-of-effect hypothesis in comparison to the effect-absence hypothesis. In contrast, when the Fun−/CA+ group was the control, the data were only 1.9 times more likely under the presence-of-effect hypothesis in comparison to the effect-absence hypothesis.

Our study did not include a Fun+/CA− group because isolated funisitis (i.e., without chorioamnionitis) is very uncommon (<5%) in preterm placentas [68,69,70]. In contrast, isolated funisitis is described in 17% of term placentas [68,69,70]. When histological examination of a placenta demonstrates both chorioamnionitis and funisitis, it is straightforward that a progressive intrauterine or intra-amniotic infectious process has occurred [70]. However, the clinical significance of isolated funisitis is less clear [70]. Potential explanations for isolated funisitis include a systemic transplacental infection, which affects the fetus but not the membranes. Alternatively, a sampling bias may occur if chorioamnionitis does not involve the entire chorioamniotic membranes [68]. In addition, isolated funisitis in term infants may occur due to damage to the cord resulting from meconium [70]. Interestingly, isolated funisitis in preterm infants is not accompanied by increased levels of cytokines in umbilical cord blood [68].

That the fetal inflammatory response induced by intrauterine infection may influence the outcome of prematurity is a biologically plausible hypothesis. The possible mechanisms linking perinatal inflammation/infection and complications of prematurity include (i) increased cardiopulmonary instability at birth with consequent higher requirement for respiratory and cardiocirculatory support and (ii) the potential pathogenic effects of hypoxia, acidosis, cytokines and other inflammatory mediators, as well as the increased oxidative stress that accompanies infection and/or inflammation [71,72,73,74]. We confirmed these higher cytokine levels associated with funisitis (see Appendix A). Although this pathophysiological situation could theoretically affect any complication of prematurity, our data suggest that IVH is the outcome most strongly influenced by the presence of a fetal inflammatory response.

IVH generally occurs within the three first days of life and affects infants with higher hemodynamic and respiratory instability, frequently associated with extreme prematurity and/or severe perinatal infections [17,75,76]. Pro-inflammatory cytokines such as IL-6, IL-1β, and tumor necrosis factor (TNF)α can induce hemodynamic disturbances through direct vascular action or by the release of vasoactive mediators, like prostacyclin and nitric oxide [74]. In addition, cytokines can induce a neuro-inflammatory cascade in the fetal brain, which may promote platelet and neutrophil activation and adhesion leading to endothelial cell damage and changes in blood rheology and flow [77,78]. These changes, occurring inside the fragile germinal matrix capillaries or within the vascular connection between germinal matrix and the subependymal venous network, may increase the risk of developing IVH [17,75,76,77,78]. However, it is noteworthy that our findings from the present meta-analysis did not encompass the more severe forms of IVH.

We have attempted to evaluate the effect of funisitis on various complications of prematurity, but it should be noted that all of these complications are multifactorial in their pathogenesis. Oxidative stress is one of the key pathogenic factors, but the causes of oxidative stress are also diverse [79,80,81,82,83]. Low GA is the main risk factor for any complication of prematurity. The development of antioxidant defenses is inversely proportional to GA, but, in addition, the more preterm infants are also the sickest. Therefore, they will have a greater need for aggressive mechanical ventilation and/or oxygen supplementation due to their more severe respiratory insufficiency [79,80,81,82,83]. Moreover, many of these infants will likely experience episodic hypoxia due to their underlying lung disease and clinical instability. They will also be more frequently exposed to postnatal infections, which increase the levels of reactive oxygen species (ROS), and to transfusions of adult erythrocytes. The latter increase the capacity of supplying oxygen to the tissues and augment the risk of iron overload and ROS generation through the Fenton reaction [79,80,81,82,83]. Additional sources of oxidative stress are parenteral nutrition [84] or phototherapy, which reduces the potential antioxidant capacity of bilirubin [85].

As mentioned above, GA is the main prognostic factor in preterm birth and as it decreases, mortality and morbidity increase. However, there is a growing awareness that, beyond GA, the pathophysiological pathway leading to prematurity plays a very relevant role in the outcome of prematurity [86,87,88,89]. The two main pathophysiological pathways, also termed endotypes, leading to very and extremely preterm birth are infection/inflammation and placental dysfunction [86,87,88,89]. Chorioamnionitis is the prototypical example of the infectious/inflammatory endotype. Numerous meta-analyses have demonstrated a robust association between chorioamnionitis and complications of prematurity, including BPD [15,90], ROP [16,91], IVH [17], PVL [92,93], NEC [94], EOS [18], LOS [18], and PDA [19,95]. However, for most of these outcomes, with the exception of IVH [17] and EOS [18], the effect of chorioamnionitis was strongly related to the fact that infants in the control group had significantly higher GAs than those in the chorioamnionitis group [15,16,17,18,19]. It is a well-known fact that the incidence of chorioamnionitis increases as GA decreases, and therefore the majority of extremely preterm newborns belong to the infectious/inflammatory endotype [7].

Our present data further suggest that fetal invasion (i.e., funisitis) occurs more frequently at lower GAs. Therefore, an undetermined component of the pathologic effect of funisitis may be related to this higher degree of prematurity rather than to the inflammatory stress that generates in the fetus. Nevertheless, while in our previous meta-analysis on chorioamnionitis [15,16,17,18,19] meta-regression showed a highly significant correlation between the lower GA of the chorioamnionitis group and the various complications of prematurity, this result could not be reproduced for funisitis (see Appendix A). However, it should be noted that the number of studies that could be included in the meta-regression was quite limited. The recommendation is to conduct meta-regression studies from a minimum of 10 studies per examined covariate [96,97] and our meta-regressions were just above that limit.

In addition to GA, infants with funisitis differ from those without funisitis in characteristics that may be critically relevant to the prognosis of prematurity, such as sex, exposure to antenatal corticosteroids, mode of birth, or exposure to other pregnancy complications. Some of these findings are not surprising since pregnancies complicated by the infectious/inflammatory endotype are less likely to be associated with fetal growth retardation, hypertensive disorders of pregnancy, or birth by cesarean section than pregnancies complicated by the placental dysfunction endotype [87,88,89,98]. However, an interesting finding of our meta-analysis is that funisitis was negatively associated with male sex. Although some studies had reported such sex differences [58,99], the finding was not consistent and there are cohorts in the literature in which male sex was associated with increased risk of funisitis as compared with female sex [100]. Interestingly, evidence from pre-clinical and clinical studies suggests that the male fetus exists in a relatively more pro-inflammatory environment than the female fetus [101]. As Challis et al., have pointed out, there is a growing need to consider fetal sex in studies of placental function and pathology [101]. In a recent meta-analysis, in which we investigated sex differences in pregnancy complications and outcomes of very preterm birth, we observed no difference in chorioamnionitis risk depending on fetal sex [102]. In contrast, hypertensive disorders of pregnancy were more frequent when the fetus was female [102]. There is a large body of evidence showing that boys are more susceptible than girls to adverse outcomes of prematurity, including BPD, ROP, NEC, IVH, chronic neurodevelopmental and cognitive impairment, and death [102,103,104]. Our group is currently conducting a meta-analysis focused on evaluating the possible effects of fetal sex on the different endotypes of prematurity and how these potential sex differences contribute to the male-female differences in prematurity outcome.

Antenatal corticosteroids in case of anticipated preterm delivery reduce infant mortality and morbidity and are the standard of care in current perinatal practice [105]. Our meta-analysis shows that the rate of use of antenatal corticosteroids is higher in preterm infants with funisitis when compared with infants who had neither funisitis nor chorioamnionitis (Fun−/CA− group). In a previous meta-analysis, we already observed that chorioamnionitis was associated with a higher rate of antenatal corticosteroid use. Paradoxically, clinical chorioamnionitis was considered for a time as a contraindication, at least relative, for the use of antenatal corticosteroids [106,107]. However, there is strong evidence that the positive effect of antenatal corticosteroids also applies to preterm infants with chorioamnionitis [108,109]. We speculate that the more frequent use of antenatal corticosteroids in the infectious/inflammatory endotype may be related to the availability of more time to prepare the fetus for preterm birth compared to the time available, for example, in hypertensive disorders of pregnancy that more often require an emergency cesarean section.

Our meta-analysis has several limitations that should be taken into account. First, as mentioned above, the number of studies for many of the outcomes analyzed was low and some of the meta-analyses had moderate or high heterogeneity. We attempted to overcome these limitations by means of the BMA meta-analysis. In addition, only three studies [47,56,61] reported a gradation of funisitis. Moreover, several studies did not report a clear definition of some of the outcomes, which made classification difficult. On the other hand, the main strength of the present study is the use of rigorous methods, including an extensive and comprehensive search and meta-analysis of baseline and secondary characteristics of the infants included in the studies.

## 5. Conclusions

In conclusion, our data suggest that funisitis is associated with an increased risk of relevant complications of prematurity, such as BPD, ROP, IVH, PVL, EOS, or death before discharge. However, infants with funisitis are also younger than infants without funisitis and part of the increased risk of complications may be related to this higher degree of prematurity. Moreover, with the exception of IVH, the presence of a fetal inflammatory response does not appear to add additional risk of complications when compared to chorioamniotic membrane inflammation without fetal involvement.

## Figures and Tables

**Figure 1 antioxidants-12-00534-f001:**
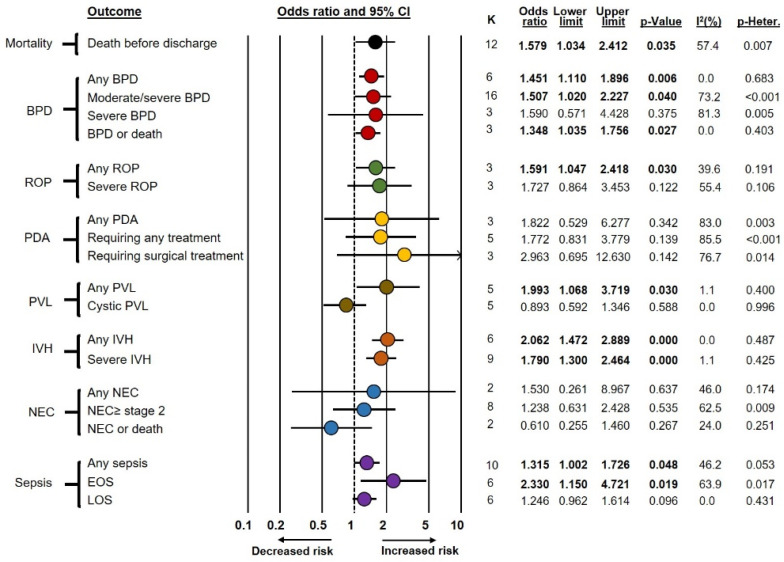
Summary of frequentist random-effects meta-analyses on the association between funisitis and short-term outcomes of prematurity. BPD: bronchopulmonary dysplasia; EOS: early-onset sepsis; IVH: intraventricular hemorrhage; K: number of studies included in the meta-analysis; LOS: late-onset sepsis; NEC: necrotizing enterocolitis; PDA: patent ductus arteriosus; PVL: periventricular leukomalacia; ROP: retinopathy of prematurity.

**Table 1 antioxidants-12-00534-t001:** Bayesian model average (BMA) meta-analysis of the association between funisitis and outcomes of prematurity.

Outcome	Control Group	K	Averaged Effect (logOR)	Standard Error	Credible Interval		BF_10_	Evidence For		*p*-Value Frequentist Analysis	BF_rf_	Evidence For	
					Lower Limit	Upper Limit		H_1_	H_0_			Random Effects	Fixed Effects
Mortality	Fun−	12	0.417	0.216	−0.048	0.823	2.008	Weak		0.035	10.65	Strong	
	Fun−/CA−	12	0.551	0.211	0.126	0.984	5.593	Mod.		0.013	6.169	Mod.	
	Fun−/CA+	12	0.118	0.201	−0.275	0.521	0.283		Mod.	0.534	0.972		Weak
Any BPD	Fun−	6	0.354	0.158	0.038	0.657	2.794	Weak		0.006	0.569		Weak
	Fun−/CA−	7	0.540	0.168	0.193	0.856	12.59	Strong		<0.001	0.706		Weak
	Fun−/CA+	7	0.175	0.186	−0.187	0.545	0.356		Weak	0.290	0.458		Weak
Mod./ severe BPD	Fun−	16	0.381	0.221	−0.055	0.828	1.272	Weak		0.040	11,651	Strong	
	Fun−/CA−	14	0.390	0.238	−0.097	0.856	1.273	Weak		0.048	42,766	Strong	
	Fun−/CA+	14	0.202	0.179	−0.154	0.570	0.472		Weak	0.262	2.078	Weak	
Severe BPD	Fun−	3	0.275	0.388	−0.442	1.109	0.508		Weak	0.375	4.333	Mod.	
	Fun−/CA−	3	0.242	0.448	−0.602	1.187	0.548		Weak	0.436	8.420	Mod.	
	Fun−/CA+	3	0.257	0.279	−0.284	0.815	0.530		Weak	0.250	0.530		Weak
BPD or death	Fun−	3	0.229	0.245	−0.374	0.625	0.819		Weak	0.027	1.293	Weak	
	Fun−/CA−	3	0.266	0.268	−0.414	0.671	1.013	Weak		0.197	1.847	Weak	
	Fun−/CA+	3	0.004	0.238	−0.485	0.456	0.241		Mod.	0.932	0.307		Mod.
Any ROP	Fun−	3	0.471	0.243	−0.130	0.849	2.199	Weak		0.030	1.660	Weak	
	Fun−/CA−	4	0.521	0.193	0.052	0.835	4.229	Mod.		<0.001	0.938		Weak
	Fun−/CA+	3	0.285	0.248	−0.217	0.747	0.677		Weak	0.113	0.644		Weak
Severe ROP	Fun−	3	0.522	0.325	−0.254	1.058	2.200	Weak		0.122	1.674	Weak	
	Fun−/CA−	3	0.527	0.361	−0.333	1.123	1.831	Weak		0.211	2.479	Weak	
	Fun−/CA+	3	0.169	0.341	−0.505	0.839	0.462		Weak	0.511	0.544		Weak
Any PDA	Fun−	3	0.415	0.440	−0.517	1.277	0.983		Weak	0.342	15.09	Strong	
	Fun−/CA−	3	0.427	0.443	−0.548	1.250	1.095	Weak		0.312	8.112	Mod.	
	Fun−/CA+	3	0.303	0.444	−0.562	1.241	0.584		Weak	0.457	2.668	Weak	
PDA req treatment	Fun−	5	0.442	0.330	−0.203	1.124	1.119	Weak		0.139	4551	Strong	
	Fun−/CA−	4	0.462	0.341	−0.238	1.131	1.210	Weak		0.158	9344	Strong	
	Fun−/CA+	4	0.006	0.230	−0.434	0.491	0.242		Mod.	0.939	0.784		Weak
PDA req surgical treatment	Fun−	3	0.675	0.484	−0.358	1.603	2.069	Weak		0.142	3.299	Mod.	
	Fun−/CA−	3	0.837	0.459	−0.111	1.713	3.489	Mod.		0.072	1.986	Weak	
	Fun−/CA+	3	0.235	0.448	−0.608	1.128	0.589		Weak	0.460	0.862		Weak
Any PVL	Fun−	5	0.585	0.327	−0.057	1.228	2.196	Weak		0.030	0.727		Weak
	Fun−/CA−	4	0.273	0.331	−0.365	0.933	0.557		Weak	0.560	0.662		Weak
	Fun−/CA+	3	0.388	0.541	−0.649	1.485	0.821		Weak	0.316	0.708		Weak
Cystic PVL	Fun−	5	−0.101	0.225	−0.541	0.344	0.296		Mod.	0.588	0.303		Mod.
	Fun−/CA−	4	−0.044	0.219	−0.469	0.392	0.264		Mod.	0.961	0.325		Mod.
	Fun−/CA+	3	−0.184	0.302	−0.773	0.415	0.437		Weak	0.222	0.591		Weak
Any IVH	Fun−	6	0.680	0.190	0.295	1.044	34.25	Strong		<0.001	0.493		Weak
	Fun−/CA−	6	0.750	0.193	0.366	1.125	65.57	Strong		<0.001	0.419		Weak
	Fun−/CA+	7	0.423	0.218	−0.005	0.853	1.896	Weak		0.026	0.662		Weak
Severe IVH	Fun−	9	0.532	0.188	0.140	0.877	7.385	Mod.		<0.001	0.632		Weak
	Fun−/CA−	9	0.593	0.191	0.191	0.945	10.22	Strong		0.001	0.758		Weak
	Fun−/CA+	9	0.079	0.216	−0.339	0.504	0.275		Mod.	0.709	0.362		Weak
Any NEC	Fun−	2	0.273	0.549	−0.791	1.377	0.730		Weak	0.637	0.929		Weak
	Fun−/CA−	2	0.271	0.570	−0.843	1.407	0.743		Weak	0.641	0.946		Weak
	Fun−/CA+	3	0.065	0.545	−0.997	1.163	0.649		Weak	0.880	0.728		Weak
NEC ≥ stage 2	Fun−	8	0.159	0.285	−0.392	0.728	0.398		Weak	0.535	10.85	Strong	
	Fun−/CA−	9	0.166	0.320	−0.453	0.821	0.422		Weak	0.549	33.68	Strong	
	Fun−/CA+	8	−0.207	0.241	−0.674	0.270	0.431		Weak	0.275	0.533		Weak
NEC or death	Fun−	2	−0.390	0.406	−1.151	0.463	0.858		Weak	0.267	0.996		Weak
	Fun−/CA−	2	−0.312	0.440	−1.130	0.634	0.744		Weak	0.724	1.193	Weak	
	Fun−/CA+	2	−0.514	0.404	−1.281	0.319	1.280	Weak		0.062	0.897		Weak
Any sepsis	Fun−	10	0.260	0.144	−0.020	0.566	1.165	Weak		0.048	3.790	Mod.	
	Fun−/CA−	10	0.289	0.140	0.023	0.587	2.051	Weak		0.022	2.773	Weak	
	Fun−/CA+	10	0.303	0.211	−0.070	0.773	0.866		Weak	0.091	3.090	Mod.	
EOS	Fun−	6	0.673	0.286	0.111	1.255	6.655	Mod.		0.019	4.281	Mod.	
	Fun−/CA−	4	0.641	0.295	0.055	1.245	4.096	Mod.		0.028	2.249	Weak	
	Fun−/CA+	4	0.179	0.335	−0.453	0.880	0.426		Weak	0.449	1.102	Weak	
LOS	Fun−	6	0.212	0.152	−0.086	0.511	0.553		Weak	0.096	0.466		Weak
	Fun−/CA−	5	0.252	0.210	−0.164	0.671	0.589		Weak	0.220	0.837		Weak
	Fun−/CA+	5	0.046	0.224	−0.390	0.493	0.269		Mod.	0.840	0.401		Weak

BPD: bronchopulmonary dysplasia; EOS: early-onset sepsis; Fun−/CA−: absence of funisitis and chorioamnionitis; Fun−/CA+: absence of funisitis but presence of chorioamnionitis; IVH: intraventricular hemorrhage; K: number of studies; LOS: late-onset sepsis; Mod.: moderate; NEC: necrotizing enterocolitis; OR: odds ratio; PDA: patent ductus arteriosus; PVL: periventricular leukomalacia; Req.: requiring; ROP: retinopathy of prematurity.

**Table 2 antioxidants-12-00534-t002:** Meta-analyses comparing infants with funisitis vs. infants who had neither funisitis nor chorioamnionitis (Fun−/CA−) or infants without funisitis but with chorioamnionitis (Fun−/CA+).

Outcome	Control Group	K	OR	95% CI		*p*	Heterogeneity		Meta-Regression *
				Lower Limit	Upper Limit		I^2^ (%)	*p*	*p*
Mortality	Fun−/CA−	12	1.162	0.721	1.874	0.537	40.3	0.072	0.179
	Fun−/CA+	12	1.840	1.163	2.911	0.009	59.8	0.004	
Any BPD	Fun−/CA−	7	1.778	1.311	2.411	<0.001	16.7	0.303	0.065
	Fun−/CA+	7	1.192	0.861	1.652	0.290	0.0	0.627	
Moderate/severe BPD	Fun−/CA−	14	1.557	1.003	2.416	0.048	76.3	<0.001	0.493
	Fun−/CA+	14	1.258	0.842	1.879	0.262	55.9	0.006	
Severe BPD	Fun−/CA−	3	1.669	0.460	6.056	0.436	83.7	0.002	0.915
	Fun−/CA+	3	1.322	0.822	2.124	0.250	0.0	0.497	
BPD or death	Fun−/CA−	3	1.324	0.864	2.029	0.197	22.5	0.275	0.121
	Fun−/CA+	3	1.015	0.719	1.433	0.932	0.0	0.950	
Any ROP	Fun−/CA−	4	1.744	1.313	2.316	<0.001	10.1	0.343	0.289
	Fun−/CA+	3	1.391	0.925	2.092	0.113	0.0	0.541	
Severe ROP	Fun−/CA−	3	1.694	0.729	3.936	0.221	66.8	0.049	0.457
	Fun−/CA+	3	1.249	0.644	2.422	0.511	0.0	0.597	
Any PDA	Fun−/CA−	3	1.850	0.561	6.102	0.312	80.1	0.007	0.919
	Fun−/CA+	3	1.688	0.425	6.705	0.457	76.5	0.014	
PDA requiring any treatment	Fun−/CA−	4	1.818	0.792	4.174	0.158	88.9	<0.001	0.308
	Fun−/CA+	4	1.018	0.639	1.623	0.939	43.9	0.148	
PDA requiring surgical treatment	Fun−/CA−	3	3.468	0.895	13.439	0.072	68.7	0.041	0.492
	Fun−/CA+	3	1.724	0.407	7.303	0.460	58.2	0.091	
Any PVL	Fun−/CA−	5	1.438	0.653	3.167	0.367	29.0	0.228	0.507
	Fun−/CA+	3	1.974	0.522	7.468	0.316	0.0	0.998	
Cystic PVL	Fun−/CA−	6	0.945	0.632	1.413	0.784	0.0	0.846	0.646
	Fun−/CA+	4	0.810	0.478	1.371	0.432	0.0	0.422	
Any IVH	Fun−/CA−	6	2.223	1.566	3.155	<0.001	0.000	0.627	0.210
	Fun−/CA+	7	1.580	1.056	2.364	0.026	0.000	0.451	
Severe IVH	Fun−/CA−	9	1.850	1.280	2.675	0.001	17.68	0.285	0.029
	Fun−/CA+	9	1.082	0.717	1.632	0.709	0.000	0.695	
Any NEC	Fun−/CA−	2	1.592	0.226	11.222	0.641	50.66	0.155	0.678
	Fun−/CA+	3	1.113	0.277	4.477	0.880	0.000	0.699	
Severe NEC	Fun−/CA−	9	1.259	0.593	2.674	0.549	68.71	0.001	0.405
	Fun−/CA+	8	0.772	0.485	1.228	0.275	0.000	0.704	
NEC or death	Fun−/CA−	2	0.778	0.194	3.125	0.724	57.65	0.124	0.701
	Fun−/CA+	2	0.499	0.240	1.037	0.062	0.000	0.617	
Any sepsis	Fun−/CA−	10	1.363	1.047	1.774	0.022	42.05	0.077	0.991
	Fun−/CA+	10	1.435	0.943	2.184	0.091	58.61	0.010	
Early-onset sepsis	Fun−/CA−	4	2.153	1.085	4.270	0.028	60.88	0.053	0.400
	Fun−/CA+	4	1.383	0.597	3.203	0.449	55.49	0.081	
Late-onset sepsis	Fun−/CA−	5	1.315	0.849	2.036	0.220	33.92	0.195	0.458
	Fun−/CA+	5	1.043	0.693	1.570	0.840	0.000	0.591	

* Comparison Fun−/CA− vs. Fun−/CA+ by meta-regresion. BPD: bronchopulmonary dysplasia; Fun−/CA−: absence of funisitis and chorioamnionitis; Fun−/CA+: absence of funisitis but presence of chorioamnionitis; IVH: intraventricular hemorrhage; K: number of studies; NEC: necrotizing enterocolitis; OR: odds ratio; PDA: patent ductus arteriosus; PVL: periventricular leukomalacia; ROP: retinopathy of prematurity.

## Data Availability

All data relevant to the study are included in the article or uploaded as Appendix A. Additional data are available upon reasonable request.

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
