# Peer review of "Association of Funisitis with Short-Term Outcomes of Prematurity: A Frequentist and Bayesian Meta-Analysis"

_antioxidants, 2023, doi:10.3390/antiox12020534_

Round 1
Reviewer 1 Report
The authors during the past decade published several papers investigating the aspects of perinatal infection and its relationship with complications of prematurity such as bronchopulmonary dysplasia (BPD), retinopathy of prematurity (ROP), necrotizing enterocolitis (NEC), intraventricular hemorrhage (IVH), periventricular leukomalacia (PVL), patent ductus arteriosus (PDA), and mortality. Chorioamnionitis represents the maternal inflammatory reaction, meanwhile, funisitis is the histologic equivalent of the fetal inflammatory response.
In the present study an attempt was made to answer this question:
“Is funisitis a risk factor for developing short-term complications (mortality, BPD, IVH, PVL, PDA, NEC, ROP, sepsis) in preterm infants compared to preterm without it?”
They conducted a systematic review and meta-analysis of studies investigating the effects of funisitis on short-term outcomes of prematurity. After a detailed search, 33 studies (12,237 infants) were included. Meta-analysis showed an association between funisitis and risk of any BPD, moderate/severe BPD, any ROP, IVH, any PVL, early-onset sepsis, and mortality. However, funisitis was not significantly associated with severe BPD, severe ROP, PDA, NEC, cystic PVL, or late-onset sepsis. When the control group was restricted to infants having chorioamnionitis without funisitis, the only outcome associated with funisitis was any IVH.
The authors also analyzed the possible differences between infants with and without funisitis for the following characteristics:
· gestational age (GA),
· birth weight (BW),
· gender,
· maternal steroid treatment.
The conclusion was that the presence of funisitis does not add an additional risk to preterm birth when compared to chorioamnionitis in the absence of fetal inflammatory response.
They also conducted a meta-regression analysis to investigate the possible correlation between the difference in GA in the presence of funisitis.
This meta-regression analysis was only performed for mortality, moderate/severe BPD, and any sepsis. None of these three meta-regressions provided statistically significant results.
As they stated this is the most extensive and comprehensive systematic review and meta-analysis that investigated the impact of funisitis on the risk of developing short-term complications in extremely preterm infants.
It is well documented that GA is the main prognostic factor in preterm infants, there is an inverse relationship between GA and mortality and morbidity indexes.
This meta-analysis shows that antenatal corticosteroid treatment rate was higher in preterm infants with funisitis compared to infants who had neither funisitis nor chorioamnionitis. In a previous meta-analysis this research group observed that chorioamnionitis was associated with a higher rate of antenatal corticosteroid treatment.
The present paper has many limitations.
1. The number of studies is low -especially after subgroups were settled-, moreover, several studies did not report a clear definition of outcomes.
2. Funisitis was detected more frequently at lower GA. Therefore, funisitis may be related to this higher degree of prematurity.
3. In addition to GA, infants with funisitis differ from those without it in characteristics that may be closely related to the prognosis of prematurity (sex, exposure to antenatal corticosteroids, mode of delivery, or complications of pregnancy).
4. The development and severity of BPD, IVH, NEC, PDA, PVL, and ROP are related to oxidative stress which is influenced by several postnatal factors, such as anemia, transfusion, bilirubin level, FiO2, mode and length of ventilation, dysnatremia, hyperglycemia, and so on.
Just a few sentences may refer to those factors, for example:
· the adult type of hemoglobin provides more oxygen to the tissues compared with the fetal type, the number of transfusions is important, iron-induced oxidative stress may play a role in the pathogenesis of oxygen radical diseases in preterm infants,
· bilirubin in low concentration serves as an antioxidant, moderately increased plasma bilirubin levels may be positively favorable to infants under oxidative stress, and phototherapy treatment may influence the bilirubin antioxidant capacity.
In summary, the research group previously provided important data related to perinatal infection and complications of prematurity, In the present study they investigated the role of fetal infection (funisitis) in the development of oxidative stress-related neonatal diseases. They hypothesized that the neonatal damage induced by the fetal inflammatory response (funisitis) is more important than the maternal infection. The data obtained after meta-analysis suggest that the presence of funisitis does not add an additional risk to preterm birth when compared to chorioamnionitis in the absence of fetal inflammatory response. However, the limitations mentioned above highlight that the diversity of the population studied may influence the statistical significance, sometimes biologically important observation is not statistically significant, or vice versa.
Author Response
The present paper has many limitations.
- The number of studies is low -especially after subgroups were settled-, moreover, several studies did not report a clear definition of outcomes.
Answer: Thank you very much for your constructive comments and suggestions for the improvement of our work. We fully agree that our study has important limitations and these are acknowledged and discussed in the manuscript. In the new version of the manuscript, we have expanded the discussion of these limitations.
The low number of studies is a very common problem in almost all meta-analyses. In fact, a study published in 2011 showed that the median number of trials included in the meta-analyses from the Cochrane Database of Systematic Reviews was 3, with an interquartile range from 2 to 6. Bayesian meta-analysis is increasingly being used to address the small sample challenge. In contrast to meta-analysis using frequentist statistics, Bayesian meta-analysis has the ability to quantify evidence in favor or against any hypothesis (including the null hypothesis), and to discriminate between absence of evidence and evidence of absence.
In this new version of our study, we have maintained the frequentist meta-analysis but in addition, we have conducted a Bayesian meta-analysis. For this, we have collaborated with a new author (František Bartoš) who has joined the study. He used our data to conduct a Bayesian model-averaged (BMA) meta-analysis. In BMA meta-analysis, multiple models are considered simultaneously and inference is proportioned to the support that each model receives from the data. This eliminates the need for selection a single preferred model (fixed-effect or random-effects model) and then interprets the model parameters without acknowledging the uncertainty inherent in the model selection. Using BMA, we estimated Bayes factors (BFs), which are the Bayesian way of quantifying results. The BF10 is the ratio of the probability of the data under the alternative hypothesis (H1) over the probability of the data under the null hypothesis (H0). The BFrf is the ratio of the probability of the data under the random effects model over the probability of the data under the fixed effect model. All this new information as well as the new methods and results are now in the new version of the manuscript. Also the discussion has been rewritten to accommodate all the new information. The following paragraph has been included:
“The main limitation of our meta-analysis is the low number of studies reporting useful data to answer the research question. This is a very common problem in me-ta-analysis. In fact, a study published in 2011 showed that the median number of trials included in the meta-analyses from the Cochrane Database of Systematic Reviews was three, with an interquartile range from two to six [64]. Bayesian meta-analysis is increasingly being used to address the small sample challenge [25, 26]. In contrast to me-ta-analysis using frequentist statistics, Bayesian meta-analysis has the ability to quantify evidence in favor or against any hypothesis (including the null hypothesis), and to dis-criminate therefore between absence of evidence and evidence of absence [25, 26, 65, 66]. This is a relevant advantage over the dichotomous interpretation of frequentist inference (significant vs. non-significant), particularly when p-values are close to the conventionally accepted limits (i.e. p<0.05) [26, 66, 67]. This is clearly illustrated in Table 1 with the ex-ample of "any IVH". When the control group consisted of infants without funisitis or chorioamnionitis (Fun-/CA-), the association between funisitis and any IVH in the frequentist analysis was "highly significant" (P<0.001). When the control group consisted of infants without funisitis but with chorioamnionitis (Fun-/CA+), the p-value (=0.026) of the frequentist meta-analysis was higher but still significant. By evaluation of BF10-values, Bayesian meta-analysis allows us to interpret that when the Fun-/CA- group was the control, the data were 65.6 times more likely under the presence of effect hypothesis in comparison to the effect absence hypothesis. In contrast, when the Fun-/CA+ group was the control, the data were only 1.9 times more likely under the presence of effect hypothesis in comparison to the effect absence hypothesis.
- Funisitis was detected more frequently at lower GA. Therefore, funisitis may be related to this higher degree of prematurity.
- In addition to GA, infants with funisitis differ from those without it in characteristics that may be closely related to the prognosis of prematurity (sex, exposure to antenatal corticosteroids, mode of delivery, or complications of pregnancy).
- The development and severity of BPD, IVH, NEC, PDA, PVL, and ROP are related to oxidative stress which is influenced by several postnatal factors, such as anemia, transfusion, bilirubin level, FiO2, mode and length of ventilation, dysnatremia, hyperglycemia, and so on.
Just a few sentences may refer to those factors, for example:
- the adult type of hemoglobin provides more oxygen to the tissues compared with the fetal type, the number of transfusions is important, iron-induced oxidative stress may play a role in the pathogenesis of oxygen radical diseases in preterm infants,
- bilirubin in low concentration serves as an antioxidant, moderately increased plasma bilirubin levels may be positively favorable to infants under oxidative stress, and phototherapy treatment may influence the bilirubin antioxidant capacity.
Answer to comments 2, 3, and 4.
We fully agree on the multifactorial character of all complications of prematurity and on the differences between the infants with and without funisitis in characteristics very relevant to the pathogenesis of these complications. We have, therefore, analyzed them in a series of additional meta-analyses and by meta-regression. Following your suggestion, we have included a new paragraph in the discussion where these differences and the multifactorial character of the complications of prematurity are clearly underlined.
“We have attempted to evaluate the effect of funisitis on various complications of prematurity but it should be noted that all of these complications are multifactorial in their pathogenesis. Oxidative stress is one of the key pathogenic factors but the causes of oxidative stress are also diverse [79-83]. Low GA is the main risk factor for any complication of prematurity. The development of antioxidant defenses is inversely proportional to GA but, in addition, the more preterm infants are also the sickest. Therefore, they will have a greater need for aggressive mechanical ventilation and/or oxygen supplementation due to their more severe respiratory insufficiency [79-83]. Moreover, many of these infants will likely experience episodic hypoxia due to their underlying lung disease and clinical in-stability. They will also be more frequently exposed to postnatal infections, which increase the levels of reactive oxygen species (ROS), and to transfusions of adult erythrocytes. The latter increases the capacity of supplying oxygen to the tissues and augments the risk of iron overload and ROS generation through the Fenton reaction [79-83]. Additional sources of oxidative stress are parenteral nutrition [84] or phototherapy, which reduces the potential antioxidant capacity of bilirubin [85].”
Reviewer 2 Report
General comments
This is an original article which aimed to analyze the hypothesis that the neonatal damage induced by the fetal inflammatory response (funisitis) is more severe than that produced when the inflammatory process is limited to the mother (chorioamnionitis without funisitis). The authors conducted a systematic review and meta-analysis using 33 studies with 12,237 infants to investigate the hypothesis. They showed an association between funisitis and risk of any BPD, moderate/severe BPD, any ROP, IVH, any PVL, early onset sepsis, and mortality. However, funisitis was not significantly associated with severe BPD, severe ROP, PDA, NEC, cystic PVL, or late onset sepsis. It is intriguing that the comorbidities of the neonates with/without funisitis or chorioamnionitis are different. However, there are several issues which should be addressed for the publication. First, the definitions of funisitis and/or chorioamnionitis are not clear. Second, statistical analysis is not enough explained. Third, there are several typos or grammatical mistakes in the manuscript.
Major comments
-
Although the authors described in the discussion section (for limitation), the definitions of funisitis and/or chorioamnionitis are not clear. These definitions are associated with the central question of this study. In addition, there are grading scores of the severity of CA or funisitis. Please add to the analysis.
-
How were the laboratory findings of CA or funisitis such as inflammatory markers or hypercytokinemia?
-
The factors shown in Table 2 could be confounding factors. Did the authors perform ANCOVA analysis to adjust the confounders?
-
Please describe more details of statistical analysis.
-
Please add the actual numbers on Figure 2 as the authors did in Figure 1.
-
Why did the authors not divide FUN+ groups into CA+ and CA-?
-
The authors defined moderate/severe BPD as oxygen requirement at the postmenstrual age of 36 weeks, and severe BPD as requiring >30% oxygen or mechanical ventilation at the postmenstrual age of 36 weeks. Please add the oxygen fraction in moderate/severe BPD.
Author Response
Major comments
- Although the authors described in the discussion section (for limitation), the definitions of funisitis and/or chorioamnionitis are not clear. These definitions are associated with the central question of this study. In addition, there are grading scores of the severity of CA or funisitis. Please add to the analysis.
Answer: Following your suggestion, in the new version of the manuscript we have included an analysis based on the grade of funisitis. Unfortunately, only three studies reported data that could be used in this analysis (supplementary table 4). The meta-analysis showed no significant differences depending on the grade of funisitis.
- How were the laboratory findings of CA or funisitis such as inflammatory markers or hypercytokinemia?
Answer: Following your suggestion, we have also included a meta-analysis comparing interleukin-6 levels in cord blood of infants with and without funisitis. These results are now in the Supplementary Table 5 and have been discussed in the new version of the manuscript.
- The factors shown in Table 2 could be confounding factors. Did the authors perform ANCOVA analysis to adjust the confounders?
Answer: The use of ANCOVA is, in general, more appropriate when individual patient data are available and not aggregated data, as is the case in our meta-analysis. On the other hand, in the association between funisitis (or chorioamnionitis) and outcomes of prematurity, variables such as gestational age (GA) are not considered confounders but intermediate variables or colliders. The criteria for confounding are that the confounder (GA) should be casually associated with both the exposure (funisitis) and the outcome (for example, BPD) (1). Since our question is whether funisitis increases the risk of developing BPD, labeling the GA as a confounder would imply the assumption that prematurity may be causing both funisitis and BPD (1-3). This scenario is highly implausible. However, if we aimed to investigate the effect of prematurity (treating it as an exposure) on the risk of BPD, funisitis would be a confounder in the backdoor path between the exposure and the outcome and should be conditioned on (1-3). Overadjustment is the undesirable consequence of adjusting for an intermediate variable that lies on a causal pathway from exposure to outcome (1).
We prefer to use meta-regression to analyze the effect that differences, for example in GA, between the exposed group and the control group may have on the effect size of the different associations. In the new version of the manuscript, we have included not only GA but also the use of antenatal corticosteroids and sex as factors analyzed by meta-regression. These new results are now included in the new version of the manuscript. In addition, and following your suggestion, we have included a more detailed description of the statistical methods related to the meta-regression and the other analyses.
- Please describe more details of statistical analysis.
Answer: The description of the statistical analysis is more detailed in the new version of the manuscript. In the new version of the manuscript, we have also included a Bayesian model-average (BMA) meta-analysis. This new method is also explained in detail.
- Please add the actual numbers on Figure 2 as the authors did in Figure 1.
Answer: The numbers are in table 2. Figure is already very complicated since two comparisons are represented (Fun+ vs Fun-/CA+ and Fun+ vs Fun-/CA-). The addition of the numbers would make the Figure extremely busy. We believe it is clearer to keep the numbers in Table 2. In the new version of the manuscript it is specified that the OR, confidence interval and heterogeneity values are depicted in Table 2.
- Why did the authors not divide FUN+ groups into CA+ and CA-?
Answer: The explanation for that is depicted in the following paragraph that has been included in the new version of the manuscript:
"Our study did not include a Fun+/CA- group because isolated funisitis (i.e. without chorioamnionitis) is very uncommon (<5%) in preterm placentas [68-70]. In contrast, isolated funisitis is described in 17% of term placentas [68-70]. When histological examination of a placenta demonstrates both chorioamnionitis and funisitis, it is straightforward that a progressive intrauterine or intra-amniotic infectious process has occurred [70]. However, the clinical significance of isolated funisitis is less clear [70]. Potential explanations for isolated funisitis include a systemic transplacental infection, which affects the fetus but not the membranes. Alternatively, a sampling bias may occur if chorioamnionitis does not involve the entire chorioamniotic membranes [68]. In addition, isolated funisitis in term infants may occur due to damage to the cord resulting from meconium [70]. Interestingly, isolated funisitis in preterm infants is not accompanied by increased levels of cytokines in umbilical cord blood [68]."
- The authors defined moderate/severe BPD as oxygen requirement at the postmenstrual age of 36 weeks, and severe BPD as requiring >30% oxygen or mechanical ventilation at the postmenstrual age of 36 weeks. Please add the oxygen fraction in moderate/severe BPD.
Oxygen fraction has been added.
References
- Ananth CV, Schisterman EF. Confounding, causality, and confusion: the role of intermediate variables in interpreting observational studies in obstetrics. American journal of obstetrics and gynecology. 2017;217(2):167-75.
- Williams TC, Bach CC, Matthiesen NB, Henriksen TB, Gagliardi L. Directed acyclic graphs: a tool for causal studies in paediatrics. Pediatric research. 2018;84(4):487-93.
- Wilcox AJ, Weinberg CR, Basso O. On the pitfalls of adjusting for gestational age at birth. American journal of epidemiology. 2011;174(9):1062-8.
Reviewer 3 Report
The article: „Association of funisitis with short-term outcomes of prematurity: A systematic review and meta-analysis“ is excellently presented review of metaanalysis of the chronic and permanent perinatal problem. The presenters presented the given objectives of the work in an adequate way, with excellent graphic, pictorial and numerical representations, and comments in the components of the article, along with a discussion of similar research and citations of adequate fresh literature. In this form, I have no objections to the components of the work, and I suggest it for publication in a journal.
Author Response
Thank you very much for your positive evaluation of our work.
Round 2
Reviewer 2 Report
Thank you for revising the manuscript. The authors answered all the questions raised by the reviewer.